# *Lactobacillus reuteri* Ameliorates Intestinal Inflammation and Modulates Gut Microbiota and Metabolic Disorders in Dextran Sulfate Sodium-Induced Colitis in Mice

**DOI:** 10.3390/nu12082298

**Published:** 2020-07-31

**Authors:** Gang Wang, Shuo Huang, Shuang Cai, Haitao Yu, Yuming Wang, Xiangfang Zeng, Shiyan Qiao

**Affiliations:** 1State Key Laboratory of Animal Nutrition, College of Animal Science and Technology, China Agricultural University, Beijing 100193, China; crazygang@126.com (G.W.); shuo0908@163.com (S.H.); c_caishuang@163.com (S.C.); 15600660793@163.com (H.Y.); wudixiaoming@163.com (Y.W.); ziyangzxf@163.com (X.Z.); 2Beijing Key Laboratory of Biological Feed Additive, China Agricultural University, Beijing 100193, China

**Keywords:** DSS-induced colitis, gut microbiota, IBD, *Lactobacillus reuteri*, metabolites

## Abstract

*Lactobacillus reuteri*, a commensal intestinal bacteria, has various health benefits including the regulation of immunity and intestinal microbiota. We examined whether *L. reuteri* I5007 could protect mice against colitis in ameliorating inflammation, modulating microbiota, and metabolic composition. In vitro, HT-29 cells were cultured with *L. reuteri* I5007 or lipopolysaccharide treatment under three different conditions, i.e., pre-, co- (simultaneous), and posttreatment. Pretreatment with *L. reuteri* I5007 effectively relieves inflammation in HT-29 cells challenged with lipopolysaccharide. In vivo, mice were given *L. reuteri* I5007 by gavage throughout the study, starting one week prior to dextran sulfate sodium (DSS) treatment for one week followed by two days without DSS. *L. reuteri* I5007 improved DSS-induced colitis, which was confirmed by reduced weight loss, colon length shortening, and histopathological damage, restored the mucus layer, as well as reduced pro-inflammatory cytokines levels. Analysis of 16S rDNA sequences and metabolome demonstrates that *L. reuteri* I5007 significantly alters colonic microbiota and metabolic structural and functional composition. Overall, the results demonstrate that *L. reuteri* I5007 pretreatment could effectively alleviate intestinal inflammation by regulating immune responses and altering the composition of gut microbiota structure and function, as well as improving metabolic disorders in mice with colitis.

## 1. Introduction

The incidence and prevalence of inflammatory bowel disease (IBD), which includes Crohn’s disease (CD) and ulcerative colitis (UC), has steadily increased over time in recent decades, especially in many newly industrialized countries [1,2]. Although the precise etiology of IBD remains unknown, there are some cases reporting that environmental factors and genetic susceptibilities contribute to dysfunctions of the intestinal epithelial barrier, and this is associated with dysbiosis of the gut microbiota and mucosal inflammation [3].

Accumulating evidence has shown that microbial dysbiosis plays a crucial role in the pathogenesis of IBD by affecting the metabolic pathway, intestinal barrier, and gut immunity [4]. Significant differences, such as reduced diversity and bacterial load, have been observed between the microbiota of patients with IBD and those of healthy individuals [5,6]. The relative abundance of several beneficial bacterial species, such as *Bacteroides*, *Eubacterium*, *Bifidobacterium*, *Faecalibacterium,* and *Lactobacillus*, have been shown to be decreased while that of Proteobacteria has been shown to increase in patients with IBD [7]. Beneficial bacterial species may protect the host mucosa from unwarranted and potentially harmful inflammatory responses [7,8]. For example, probiotic combinations including *Lactobacilli* and *Bifidobacteria* can suppress inflammation when administered to patients [9]. Therefore, the therapeutic modulation of imbalanced gut microbiota has been widely used for the treatment of IBD and has been proven to be a promising approach to therapy with significant clinical efficacy [10].

Probiotics are defined as living commensal microorganisms which have a health benefit on the host when administrated in adequate amounts that are important to intestinal health when consumed [11] and have been shown to ameliorate inflammation in experimental models, as well as in clinical trials [12,13,14]. The mechanisms of probiotics on gut health include intestinal barrier function modulation, the production of bacteriocin or antibacterial proteins, and the inhibition of pathogens by competing for common receptors of adhesion [15]. *Lactobacillus reuteri* is a commensal intestinal species in the gastrointestinal tract of humans and animals [16]. Numerous studies have demonstrated that *L. reuteri* can suppress proinflammatory cytokines in intestinal epithelial cells [17] and monocytes [18] and intestinal inflammation in different rodent models [19,20]. In our previous work, *L. reuteri* I5007 was isolated from the colonic mucosa of healthy weaning piglets [20]. Several studies have shown that *L. reuteri* I5007 has several probiotic properties including strong adhesion in the intestine, competitive rejection pathogens, and modulation of immune function [21,22,23]. However, whether *L. reuteri* I5007 could ameliorate intestinal inflammation and injury by modulating gut microbiota and metabolic pathways in colitis models is still unknown.

Therefore, we hypothesized that *L. reuteri* I5007 could have a protective role in ameliorating intestinal inflammation. The anti-inflammatory activities, modulation of gut microbiota and metabolic pathways of *L. reuteri* I5007 were investigated in the present study. We initially examined the in vitro effect of *L. reuteri* I5007 on modulating proinflammatory cytokine expression in the HT-29 cell model. We subsequently demonstrated the effects of *L. reuteri* I5007 pretreatment on colonic inflammation, colonic microbiota composition, and metabolic pathways in a dextran sulfate sodium (DSS)-induced mouse model of colitis.

## 2. Materials and Methods

### 2.1. Ethics Statement

All animal procedures used in this study were approved by the Animal Care and Use Committee at China Agricultural University (201605510410554).

### 2.2. Preparation of Bacteria

*L. reuteri* I5007 was cultured in de Man, Rogosa, and Sharpe (MRS; Solarbio Science and Technology Co. Ltd., Beijing, China) medium at 37 °C for 20 h under anaerobic conditions. Then, the pellet was washed with phosphate-buffered saline (PBS) and adjusted to a density of 10^9^ colony forming units (CFU) mL^−1^ with a Roswell Park Memorial Institute (RPMI, Gibco, Carlsbad, CA, USA) 1640 basic medium (in vitro) or PBS (in vivo).

### 2.3. Cell Culture and Treatment

Human colon cell line HT-29 cells from the Cell Bank of the Chinese Academy of Sciences (Beijing, China) were maintained in a RPMI 1640 basic medium supplemented with 10% (*v/v*) heat-inactivated fetal bovine serum (FBS), 100 U mL^−1^ penicillin (Sigma, San Jose, CA, USA), and 100 g mL^−1^ streptomycin (Sigma, USA) at 37 °C in a humidified 5% CO_2_ and 95% air atmosphere. When cells reached 80% confluency, the complete medium was removed completely prior to the challenge study, and cells were fed with a fresh RPMI 1640 basic medium lacking antibiotics. HT-29 cells were then subjected to L. *reuteri* I5007 (10^9^ CFU mL^−1^) or lipopolysaccharide (1 µg/mL, LPS, Sigma USA, *E. coli* 0111:B4) treatment under three different conditions, i.e., pre-, co- (simultaneous), and posttreatment. For pretreatment conditions, *L. reuteri* I5007 was first added to the wells and incubated at 37 °C and 5% CO_2_ for 6 h. After incubation, *L. reuteri* I5007 was challenged with LPS and further incubated for an additional 4 h. For cotreatment, HT-29 cells were challenged with *L. reuteri* I5007 and LPS simultaneously for 10 h. Last, for posttreatment, HT-29 cells were initially challenged with LPS and incubated for 4 h. The spent medium was removed, and subsequently, the LPS-treated cells were subjected to *L. reuteri* I5007 for an additional 6 h.

### 2.4. Animals and Treatments

Forty-eight 6- to 7-week-old female C57BL/6 mice (HFK Bioscience Co., Ltd., Beijing, China) were randomly assigned to four treatments (12 mice each group): Control, I5007, DSS, and DSS_I5007 groups. From day 1 to day 14, the control and DSS groups received orally PBS (200 μL), and the others orally administered *L. reuteri* I5007 (10^9^ CFU mL^−1^) suspended in 200 μL of PBS. Colitis was induced by 3% DSS (molecular weight: 5000; MP Biomedicals) from day 8 to 14, followed by two days of drinking normal water. Mice were weighted every day and sacrificed on day 16. Once the mice were sacrificed, their colons were removed aseptically, and the length was measured. The serum, colon, and colonic contents were collected for follow-up experiments.

### 2.5. Histological and Goblet Cell Evaluation

Colonic tissues were fixed with 4% paraformaldehyde, embedded in paraffin wax, sliced, and stained with hematoxylin and eosin (H&E) or Alcian blue. Histological pathology was observed with a light microscope (Olympus XC41, Tokyo, Japan).

### 2.6. Quantitative Real-Time Polymerase Chain Reaction (PCR) Analysis

Total RNA from cells and mouse colon tissue was extracted using TRIzol reagent (Invitrogen, Carlsbad, CA) and a cryogenic lapping instrument (JXFSTPRP-CL, Shanghai, China). The isolation of total RNA was followed by a previously described method [24]. The concentration of RNA was determined with a spectrophotometer (NanoDrop 2000, Thermo Scientific Co. Ltd., Wilmington, Delaware, USA). The primers for real-time PCR are shown in Table 1. Glyceraldehyde-3-phosphate dehydrogenase (GAPDH) was used as an internal reference in this study. The relative mRNA expression of the target genes was determined using the 2^−∆∆Ct^ method.

### 2.7. Assay of Inflammatory Cytokines

Secreted cytokines in the serum were analyzed using the LEGENDplex™ Mouse Th17 Panel (8-plex) array (Biolegend, San Diego, CA, USA) according to the manufacturer’s protocol. The data were collected on an LSR Fortessa flow cytometer and analyzed using LEGENDplex™ software version 7.0 (Biolegend). The 8-plex contained the following antibodies specific for cytokines: IFN-γ, TNF-α, IL-6, IL-10, IL-17A, IL-17F, IL-22, and IL-21.

### 2.8. Bioinformatics Analysis of 16S rRNA Gene Profiling

Total DNA was extracted from the colonic contents using a QIAamp-DNA stool mini kit (Qiagen, Hilden, Germany) according to the manufacturer’s instructions. The DNA samples were used as the template for PCR amplification of the V3-V4 hypervariable regions of 16S rRNA genes by using the primers 338F (5′-GTGCCAGCMGCCGCGG-3′) and 806R (5′-CCGTCAATTCMTTTRAGTTT-3′). PCR amplification was performed on an ABI GeneAmp^®^9700 PCR system (Applied Biosystems, Foster City, CA, USA), and the PCR amplification products were quantified with a QuantiFluor™-ST Handheld Fluorometer with UV/blue channels (Promega Corporation, Madison, WI, USA). Sequencing reactions were subsequently performed using the Illumina Miseq sequencing technology (Illumina Inc., San Diego, CA, USA) for paired-end reads. Operational taxonomic units (OTUs) were delineated at the cutoff of 97% using USEARCH version 10.0 [25]. Heatmap generation and PCoA (principal coordinate analysis) were performed on the relative abundance of OTUs or Kyoto Encyclopedia of Genes and Genome (KEGG) pathways using R (version 2.15). Statistically significant differences in the relative abundance of associated taxa were determined using the linear discriminant analysis (LDA) effect size (LEfSe) [26]. Phylogenetic Investigation of Communities by Reconstruction of Unobserved STates (PICRUSt) was used to predict the altered KEGG pathways based on 16S sequencing data [27]. Predicted functional genes were categorized into KEGG pathways and compared across groups using STAMP (Statistical Analysis of Metagenomic Profiles) [28].

### 2.9. Untargeted Metabolomic Study

Metabolite concentrations in mouse colonic contents were quantified using a UPLC-TOF-MS. Fifty milligrams of samples were transferred into 1.5 mL centrifuge tubes and mixed with 400 μL of cold methanol/acetonitrile (1:1, *v/v*). The mixture was homogenized for 5 min and subsequently centrifuged at 14,000× *g* for 15 min at 4 °C.

The profiling of metabolites used the procedures described as before [29]. Multivariate statistical analysis was performed with SIMCA-P. Orthogonal projection to latent structures discriminant analysis (OPLS-DA) were performed to identify and rank signature metabolites by classification information among the groups. Only metabolites with variable importance (VIP) greater than 1 were further analyzed based on the Student’s t-test at the univariate level. Adjusted *p*-values lower than 0.05 were considered statistically significant. The metabolic KEGG pathway enrichment was used by MetaboAnalyst online (www.metaboanalyst.ca).

### 2.10. Statistical Analysis

The data are expressed as the mean ± SEM and analyzed using GraphPad Prism 6.0 (GraphPad Software, La Jolla, CA, USA). Significant differences between the two groups were evaluated by the two-tailed unpaired Student’s t-test or Mann-Whitney U tests for samples that were not normally distributed. Significant differences among three or more groups were evaluated by one-way ANOVA and Tukey’s post hoc test in the case of normal distribution, or Kruskal-Wallis with non-normal or non-parametric distribution. The level of significance was set at *p* < 0.05 (* *p* < 0.05; ** *p* < 0.01; *** *p* < 0.001; and **** *p* < 0.0001).

## 3. Results

### 3.1. Effects of L. reuteri I5007 on Inflammatory Cytokines in HT-29 Cells Challenged with LPS

Challenging HT-29 cells with lipopolysaccharides (LPS) for 4 h led to significantly increased mRNA expression of IL-8, TNF-α, and IL-1β and reduced IL-10 mRNA expression compared with the control (Figure 1). Treatment with *L. reuteri* I5007 alone did not change the expression of inflammatory cytokine mRNA. On the other hand, when HT-29 cells were pretreated, cotreated or posttreated with *L. reuteri* I5007, the expression levels of IL-8 and TNF-α were significantly downregulated compared to those challenged with LPS. However, compared to the control, the expression levels of TNF-α and IL-10 did not change in the cotreatment and posttreatment groups, respectively (Figure 1A,B). The pretreatment led to a significant upregulation of the anti-inflammatory cytokine IL-10 and a decrease in the proinflammatory cytokines IL-1β and TNF-α (Figure 1C). Hence, the most effective immunosuppressant treatment, as it was able to regulate inflammation induced by LPS in HT-29 cells, was under pretreatment conditions.

### 3.2. Effects of L. reuteri I5007 on Colitis Symptoms

The pretreatment of *L. reuteri* I5007 to mice for one week before colitis induction did not result in any sign of toxicity in comparison with untreated mice, as evaluated by body weight increase, food intake, and general appearance of the animals (data not shown). Untreated and *L. reuteri* I5007-treated mice continuously changed to stable, while DSS-treated mice began to lose weight after three days of DSS administration (Figure 2A). However, the *L. reuteri* I5007 pretreatment significantly decelerated weight loss (Figure 2A). *L. reuteri* I5007 minimized the reduction in colon length induced by DSS treatment (Figure 2B,C). Histological examination of the colon treated with DSS showed severe microscopic inflammation of the mucosa, as indicated by ulceration, edema, crypt damage, and infiltration of the intestinal epithelial layer (Figure 2D). In contrast, pretreatment with *L. reuteri* I5007 inhibited this inflammatory response (Figure 2D). The mucus layer structure almost disappeared after the DSS treatment, while *L. reuteri* I5007 could restore the numbers of goblet cells and the mRNA expression of MUC-2 (Figure 2E,F).

### 3.3. Effects of L. reuteri I5007 on Cytokines Expression in Colon Tissue and Serum

Compared with the normal control group, there was a higher mRNA expression of IL-6, IL-1β, IFN-γ, TNF-α, and IL-17A in the colon tissue of the DSS-treated group (Figure 3A). However, *L. reuteri* I5007 significantly reduced the increase in IL-6, IL-1β, TNF-α, and IL-17A expression caused by DSS (Figure 3A).

Administration of DSS increased the secretion of IL-6, IL-17A, TNF-α, and IFN-γ but reduced the level of IL-10 in the serum compared with the normal control group (Figure 3B). Pretreatment of colitis mice with *L. reuteri* I5007 significantly reduced the DSS-induced production of TNF-α, IL-6, IFN-γ, and IL-17A, as well as increased the level of IL-10 compared with the DSS group (Figure 3B). Additionally, oral pretreatment of *L. reuteri* I5007 in the normal group did not influence the production of inflammatory cytokines.

### 3.4. Effects of L. reuteri I5007 on Microbiota Composition in Colon Contents

The alpha diversity of microbial communities, as indicated by the richness index, was reduced significantly by DSS treatment, while pretreatment of *L. reuteri* I5007 did not influence the microbial alpha diversity (Figure 4A). The principal coordinate analysis (PCoA) based on Bray-Curtis dissimilarity visualized the relative similarity of microbiota composition and clearly illustrated that the normal, DSS, and DSS_I5007 groups had unique microbiota structures (Figure 4B). Figure 4C shows that the predominant bacterial communities at the phylum level in healthy mice were Bacteroidetes, Firmicutes, Proteobacteria, Tenericutes, and Actinobacteria. Treatment with DSS led to a significant increase in the relative abundance of Proteobacteria and a decrease in that of Bacteroidetes, whereas the DSS_I5007 group had a significantly decreased relative abundance of Proteobacteria and an increased Firmicutes and Bacteroidetes relative abundance (Figure 4C). Figure 4D shows the 10 most abundant bacterial families among the four groups. DSS treatment exhibited a significantly decreased abundance of *Lactobacillaceae* (1.1%) but a significantly increased *Enterobacteriaceae* abundance (17.7%) (Figure 4D). Interestingly, the abundance of *Lactobacillaceae* (1.5%) did not change, but that of *Erysipelotrichaceae* (13.6%) increased in the DSS_I5007 group (Figure 4D). In the DSS_I5007 group, the abundance of *Enterobacteriaceae* (2.8%) was reduced, and there was a similar bacterial family composition as that in the normal mice (Figure 4D).

Using LEfSe to compare gut microbiota among groups. Mice from the control group showed higher relative abundances of the bacterial families *Prevotellaceae*, *Eubacteriaceae,* and *Lactobacillaceae* and the genera *Lactobacillus*, *Eubacterium,* and *Ruminococcus* (Figure 5A). The DSS group showed an increased abundance of potential pathogenic bacteria, such as the phylum Proteobacteria, and its lower taxa family *Enterobacteriaceae* and genus *Escherichia-Shigella* (Figure 5A). The DSS_I5007 group exhibited an increase in Bifidobacteriales and Deferribacterales at the order level and *Deferribacteraceae*, *Bifidobacteriaceae*, *Peptostreptococcaceae*, *Helicobacteraceae,* and *Erysipelotrichaceae* at the family level, as well as *Romboutsia*, *Bifidobacteriales, Mucispirillum,* and *Clostridium_XIII* at the genus level (Figure 5A).

### 3.5. Effects of L. reuteri I5007 on Microbiota Functional Composition in Colonic Contents

In addition to taxonomic composition, we used Phylogenetic Investigation of Communities by Reconstruction of Unobserved STates (PICRUSt) to infer the metagenomic functional content (Figure 5B,C). Compared with the normal control, DSS mainly caused changes in some metabolic and biosynthesis pathways, such as amino acid metabolism, D-alanine metabolism, vitamin metabolism, energy metabolism, secondary bile acid biosynthesis, biosynthesis of other secondary metabolites, tetracycline biosynthesis, primary bile acid biosynthesis, and peptidoglycan biosynthesis (Figure 5B). Pretreatment of *L. reuteri* I5007 in colitis mice significantly influenced the biosynthesis and biodegradation of secondary metabolites, flavonol biosynthesis, amino acid metabolism, and antibiotic biosynthesis compared to the DSS group (Figure 5C).

### 3.6. Effects of L. reuteri I5007 Metabolite Composition in Colonic Contents

Score plots of orthogonal projections to latent structures-discriminate analysis (OPLS-DA) revealed that the metabolites in the colonic contents of the two groups (control vs. DSS and DSS vs. DSS_I5007) clustered separately (Figure 6A,B).

Sixty-four differentially expressed metabolites were identified between the control and DSS groups, and DSS administration upregulated fifty-one metabolites (Figure 6E). These metabolites were mostly organic acids and their derivatives. These differentially expressed metabolites were mapped to KEGG pathways and enriched in 10 specific pathways (Figure 6C) with *p*-values <0.05. These pathways mainly involved amino acid metabolism and biosynthesis, mineral absorption, ABC transporters, and aminoacyl−tRNA biosynthesis (Figure 6F). Compared to the DSS group, *L. reuteri* I5007 pretreatment changed thirty-five metabolites, and most were carbohydrates and their derivatives. There were eight significantly enriched KEGG pathways in the two groups and carbohydrate metabolism-related pathways, including central carbon metabolism in cancer, the pentose phosphate pathway, starch and sucrose metabolism, carbohydrate digestion and absorption, and galactose metabolism, were enriched (Figure 6D). The most significantly enriched pathway in the two groups was that of ABC transporters, which was the same as the enriched pathways between the control and DSS groups.

## 4. Discussion

In the present study, we found that pretreatment with *L. reuteri* I5007 reduced the inflammation in LPS-challenged HT-29 cells and alleviated experimental colitis in mice. Pretreatment of *L. reuteri* I5007 maintained the consistency of the colon, improved colon length, maintained body weight, and restored the numbers of globet cells. The preventive effects of *L. reuteri* I5007 were attributed to critical changes in intestinal microbiota and metabolites, as well as their roles in improving colon function. These results provided important insights into the preventive effects of probiotics on colonic inflammation.

Many studies have demonstrated the beneficial effects of probiotic strains or their combinations in the treatment or prevention of human or experimental intestinal inflammation models. For instance, TNBS-induced colitis in rats was ameliorated by both *Lactobacillus fermentum* CECT5716 and *L. reuteri* ATCC55730 [30]. A mixture of eight different strains of probiotic bacteria (VSL#3) reduced inflammation in mice with DSS-induced colitis [12]. In addition, in IL-10-deficient mice, which spontaneously develop colitis, *L. reuteri* was shown to attenuate colonic inflammation [31]. In the present study, we found that *L. reuteri* I5007 effectively mitigated body weight loss and increased colon length in DSS-induced colitis mice. Colonic histopathology changes showed that pretreatment of *L. reuteri* I5007 decreased mucosal ulceration, epithelial edema and crypt loss, as well as inflammatory cell infiltration.

Cytokines are known to be involved in the pathogenesis of IBD, which modulates intestinal mucosal inflammation, as well as epithelium integrity [32]. It has been noted that the reduction in inflammatory cytokines, such as TNF-α, IL-6, IL-1β, IFN-γ, and IL-17A, in the serum represents a target for IBD therapy, as they play leading roles in the formation of colitis [33]. The present study demonstrated the ability of *L. reuteri* I5007 to reduce inflammation in DSS-treated mice, as shown by decreased levels of the proinflammatory cytokines IFN-γ, TNF-α, IL-1β, IL-6, and IL-17A in the colon tissue and serum. Moreover, these proinflammatory cytokines (e.g., TNF-α, IL-1β, and IL-6) act as the activator of NF-κB, which can further promote the secretion of proinflammatory cytokines, and then enhance the inflammation [34]. IL-10 is a key anti-inflammatory cytokine and ameliorates the overproduction of proinflammatory cytokines [35]. These results suggest that *L. reuteri* I5007 has an important impact on systemic and intestinal anti-inflammation in mice with DSS-induced colitis. Moreover, parallel in vitro studies in an LPS-induced inflammatory cell model exhibited that *L. reuteri* I5007 had strong modulating effects on diverse inflammatory mediators.

Colonic mucus has been shown to be thinner, and mucus properties have been shown to be altered in patients with active UC, as well as in colitis mice and rats [36,37]. These changes in the pre-epithelial barrier allow bacteria to penetrate the normally impermeable inner and firmly adherent mucus layer [37,38]. IBD has been associated with mucus-producing defects and a decrease in the number of goblet cells [39]. Probiotics have previously been shown to affect the production of mucus [40]. From the results of colon tissue sections in the present study, goblet cells were completely damaged, and a large amount of infiltration of immune cells in the DSS group was observed. In agreement with the other reports, pretreatment of *L. reuteri* I5007 stimulated the expression of MUC2, increased the number of goblet cells, and restored tissue damage.

Compelling studies have revealed that broad intestinal microbial dysbiosis, including a reduction in bacterial abundance and a decrease in diversity, occurs in IBD [41]. It has been reported that probiotics could improve the clinical outcome of patients with IBD and animal models by influencing the abundance and composition of gut microbiota [42,43]. As expected, we observed that DSS treatment caused significantly decreased richness and a change in the diversity of gut microbiota compared with the control group, a finding that is consistent with that in previous studies on patients with IBD and the DSS-induced colitis model [44,45]. In the DSS_I5007 group, we observed a change in β diversity and an altered relative abundance of specific bacterial taxa. The PCoA analysis results suggest that there were distinct bacterial communities among the control normal group, DSS group, and DSS_I5007 group. However, restored α diversity in the group supplemented with *L. reuteri* I5007 was not observed in the present study, indicating that the *L. reuteri* I5007 pretreatment cannot fully reverse the gut microbiota abundance.

*L. reuteri* I5007 pretreatment dramatically decreased the abundance of Proteobacteria and increased that of Firmicutes, which were also reported in IBD patients [46,47]. Specific bacteria in the *Enterobacteriaceae* family were enriched in both mice and patients with IBD [48]. In human studies, *Escherichia* was found in patients with UC and in ileac biopsies of individuals with CD [49,50]. Similar results were observed in our study, in which *Enterobacteriaceae* was clustered in mice treated with DSS, and overgrowth of this bacterium may destroy intestinal permeability, resulting in worsened colitis. Pretreatment of *L. reuteri* I5007 decreased the abundance of the *Enterobacteriacae* family and the genus *Escherichia-Shigella*. Previous studies in experimental models and in patients with IBD have also documented a depletion in bacterial species that provide benefit to the host, such as *Bifidobacterium* and *Lactobacillus* [51]. These bacteria produce antimicrobial substances, compete with pathogens for epithelial and mucin-binding sites, and have been shown to attenuate symptoms and maintain remission of UC [52,53]. LEfSe revealed that *Bifidobacterium* was dominant in the DSS_I5007 group and might help to modulate gut inflammation. However, the pretreatment of *L. reuteri* I5007 alone in the normal mice did not modulate microbiota composition, which was similar to the normal control group. A randomized controlled trial (RCT) study reported that probiotics did not significantly impact the fecal microbiota composition of healthy subjects [54]. This indicates that probiotics may not influence a stable microbiota. Therefore, these findings provide novel evidence that *L. reuteri* I5007 is an effective modulator of intestinal microbiota dysbiosis caused by intestinal inflammation.

To understand the functional interactions between microbiota and hosts, we investigated the predicted functional differences in KEGG pathways predicted by PICRUSt. We observed that the DSS treatment disrupted the balance of amino acid metabolism, including glycine, serine, threonine, glycine, serine, threonine, taurine, glutamine, and alanine metabolism, in the colonic contents. Glycine, cysteine, and glutamate can create glutathione (GSH), an important antioxidant in the body. GSH, cysteine, and taurine have been reported to reduce inflammatory parameters in an oxidant-mediated injury rat model [55,56]. Amino acids play an important role in biological activities, providing energy and materials for many biosynthesis processes in the body. We noticed that the metabolism of several amino acids, including tyrosine, pyruvate, cysteine, methionine, glutamine, histidine, and thiamine, was upregulated by the pretreatment of *L. reuteri* I5007 in colitis mice. Much research has indicated that the metabolism of these amino acids is linked with the development of IBD and gut microbiota dysbiosis [57,58].

Changes in the composition of the gut microbiota led to metabolite changes that impact IBD pathogenesis [59]. Next, we investigated the change in metabolites in the colonic contents using UHPLC. We observed that several amino acid levels were higher in the colonic contents after DSS treatment than in the colonic contents of the control group. Similar to our findings, multiple studies have reported that amino acid concentrations were decreased in the feces and increased in the serum in patients with UC or CD [60,61]. Inflammation was alleviated in colitis with the treatment of glycine, arginine, and histidine, and the metabolites of arginine and histidine can promote the growth of muscles and body [62]. Pretreatment of *L. reuteri* I5007 increased the concentration of multiple carbohydrates. Carbohydrates can be degraded and fermented by the gut microbiota into monosaccharides, as well as various byproducts. Recent accumulating evidence has indicated that these gut microbial byproducts, such as short chain fatty acids, can modulate the host immune system and intestinal barrier function, thus playing a crucial role in intestinal homeostasis [63,64]. The downregulation of urocanic acid in the DSS_I5007 group indicated that *L. reuteri* I5007 may increase the antioxidant function and decreased oxidative stress of the host, which was an important reason for the development and continuation of UC [65]. At the pathway level, by metabolomic analyses, we observed that several KEGG pathways were enriched, such as ABC transporters and carbohydrate metabolism-related pathways (e.g., the galactose phosphate pathway, starch and sucrose metabolism, pentose phosphate pathway, and carbohydrate digestion and absorption) after treatment with *L. reuteri* I5007 in colitis mice. It is reported that the expression of ABC transporters was dysregulated in the colon of patients with IBD [66]. These metabolites and pathways were greatly adjusted by the *L. reuteri* I5007 pretreatment, indicating that *L. reuteri* I5007 may alleviate inflammation by adjusting the broken balance of amino acid metabolism and improving the transport system and energy supply.

## 5. Conclusions

In summary, the *L. reuteri* I5007 pretreatment could alleviate mucosal inflammation by microbe-host interactions that protect intestinal epithelial cells from injury and downregulate the levels of related inflammatory cytokines. Pretreatment of *L. reuteri* I5007 also modulated the microbial community and functional composition, as well as improved metabolic functions, including amino acid metabolism, vitamin metabolism, and carbohydrate metabolism, which are related to energy supply and antioxidant capacity. Further research is needed to explore the exact mechanism by how *L. reuteri* I5007 influences the interaction between the gut microbiota and host. However, our study shows that *L. reuteri* I5007 exerted a protective effect on DSS-induced colitis in mice, indicating that *L. reuteri* I5007 may be a potential probiotic agent for ameliorating colitis and a great therapeutic potential for IBD prevention.

## Figures and Tables

**Figure 1 nutrients-12-02298-f001:**
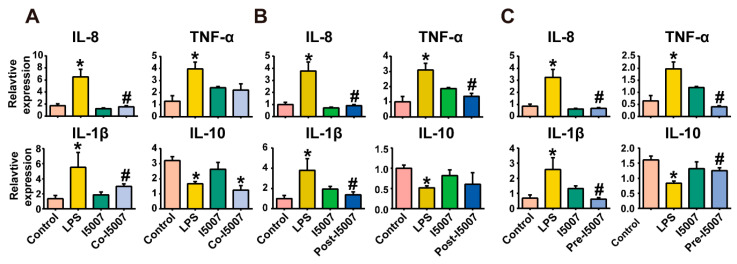
Induction of inflammatory cytokine expression in HT-29 cells upon treatment with lipopolysaccharides (LPS) and *L. reuteri* I5007. (**A**) Cotreatment. HT-29 cells were challenged with *L. reuteri* I5007 and LPS simultaneously for 10 h; (**B**) posttreatment. HT-29 cells were initially challenged with LPS for 4 h, and then L. *reuteri* I5007 was added 6 h after removal of the medium; (**C**) pretreatment. HT-29 cells were first treated with *L. reuteri* I5007 for 6 h and then challenged with LPS for 4 h. Results were compared by the one-way ANOVA and Tukey’s post hoc test. * *p* < 0.05 compared with the control group; # *p* < 0.05 compared with the LPS group.

**Figure 2 nutrients-12-02298-f002:**
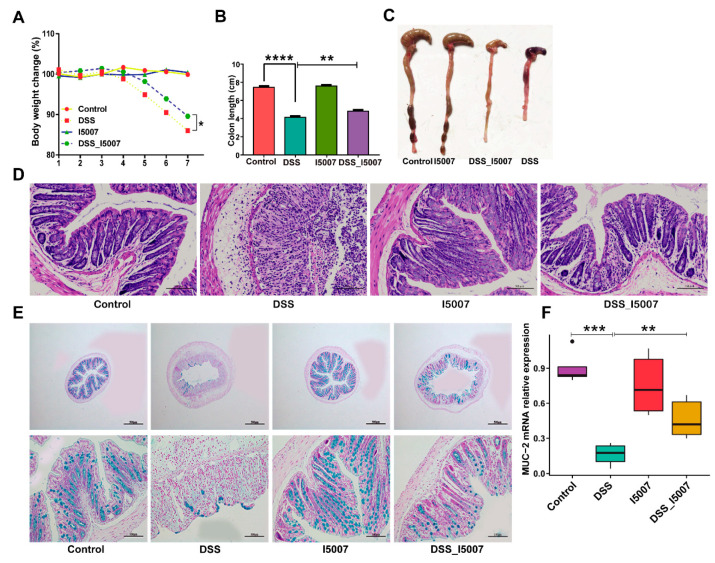
Effects of *L. reuteri* I5007 pretreatment on the severity of dextran sulfate sodium (DSS)-induced colitis. (**A**) DSS (3%) was added to drinking water for seven days, and *L. reuteri* I5007 was administered to the mice seven days before the DSS challenge until the end of colitis induction; (**B**) body weight change at the end of the experiment; (**C**) colon length; (**D**) hematoxylin and eosin (H&E) staining of colon tissue; (**E**) Alcian blue staining of goblet cells; (**F**) the expression of MUC-2 in the colon tissue. Results were compared by one-way ANOVA and Tukey’s post hoc test. * *p* < 0.05; ** *p* < 0.01; *** *p* < 0.001; **** *p* < 0.0001.

**Figure 3 nutrients-12-02298-f003:**
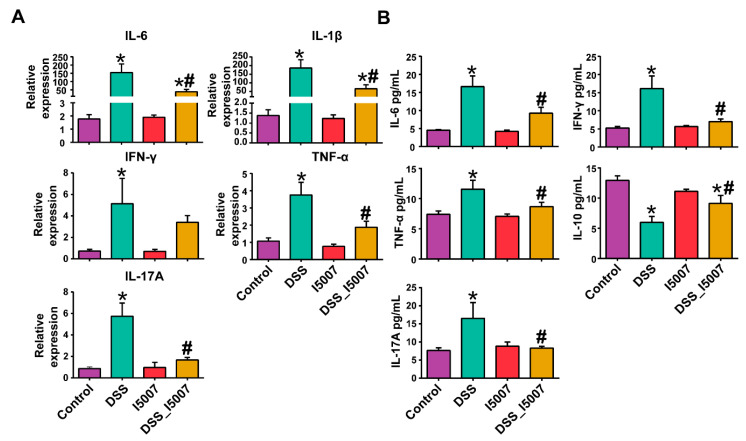
*L. reuteri* I5007 modulates the production of inflammatory cytokines in the colon tissue and serum. (**A**) Gene expression of inflammatory cytokines in the colon tissue; (**B**) the level of inflammatory cytokines in the serum. Results were compared by one-way ANOVA and Tukey’s post hoc test. * *p* < 0.05 compared with the control group; # *p* < 0.05 compared with the DSS group.

**Figure 4 nutrients-12-02298-f004:**
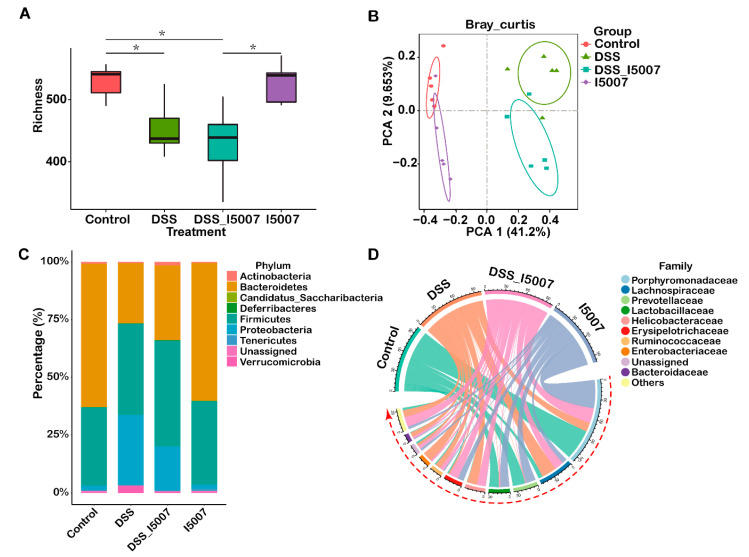
Impact of *L. reuteri* I5007 pretreatment on colonic microbiota composition. (**A**) Boxplot of microbiota alpha diversity indicated by the richness index; (**B**) principal component analysis (PCoA) plot of microbiota based on Bray-Curtis dissimilarity; (**C**) histogram of structural composition of microbiota at the phylum level; (**D**) chord diagram of the structural composition of microbiota at the family level.

**Figure 5 nutrients-12-02298-f005:**
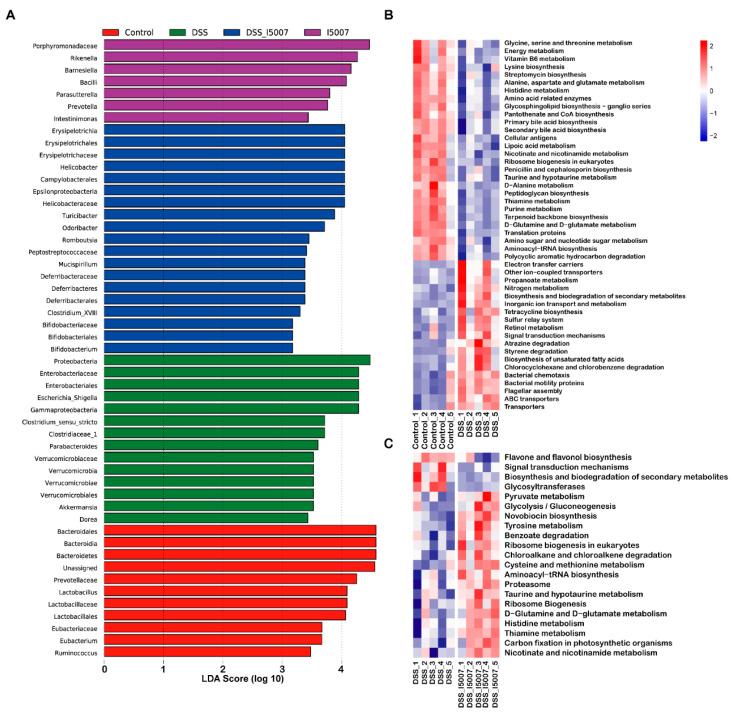
Effects of *L. reuteri* I5007 pretreatment on colonic-specific bacterial taxa and microbiota functional composition. (**A**) Histogram of the linear discriminant analysis (LDA) results for the differentially expressed features; (**B**) heatmap of significantly enriched Kyoto Encyclopedia of Genes and Genome (KEGG) pathways between the control and DSS groups; (**C**) heatmap of significantly enriched KEGG pathways between the DSS and DSS_I5007 groups. The significance of KEGG pathways compared across groups using Statistical Analysis of Metagenomic Profiles.

**Figure 6 nutrients-12-02298-f006:**
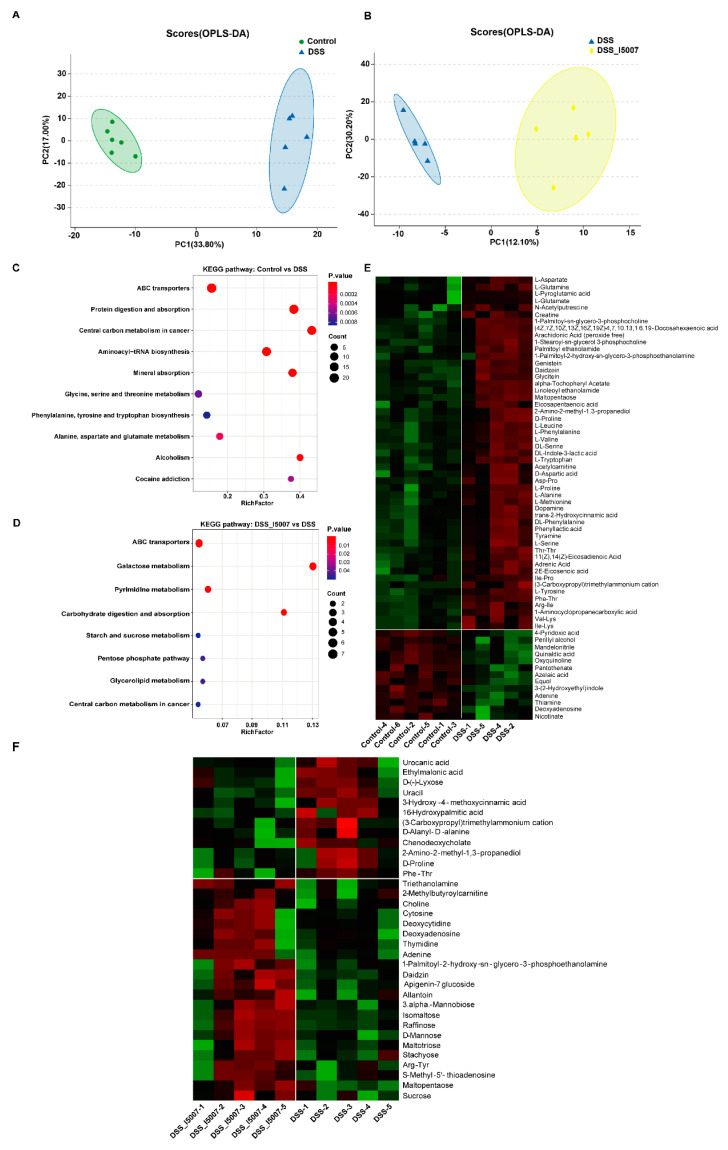
Effects of *L. reuteri* I5007 on the metabolite composition in colonic contents. (**A**) Orthogonal projections to latent structures-discriminate analysis (OPLS-DA) score plots between the control and DSS groups; (**B**) OPLS-DA score plots between the DSS_I5007 and DSS groups; (**C**) KEGG pathways analysis presented by bubble plot between the control and DSS groups; (**D**) KEGG pathway analysis between the DSS_I5007 and DSS groups presented as a bubble plot; (**E**) heatmap of the differentially expressed metabolites in the colonic contents between the control and DSS groups; (**F**) heatmap of the differentially expressed metabolites in the colonic contents between the DSS_I5007 and DSS groups. The significance of KEGG pathways compared across groups using STAMP.

**Table 1 nutrients-12-02298-t001:** Primer sequences used for real-time RT-PCR.

Target Genes	Forward (F)/Reverse (R)	Sequence	Size (bp)
M-GAPDH	F	AACTTTGGCATTGTGGAAGG	115
	R	ACACATTGGGGGTAGGAACA	
M-IL-6	F	AGACTTCCATCCAGTTGCCT	175
	R	CATTTCCACGATTTCCCAGAGA	
M-IL-1β	F	TCAGGCAGGCAGTATCACTC	250
	R	AGCTCATATGGGTCCGACAG	
M-IFN-γ	F	ACTGGCAAAAGGATGGTGAC	237
	R	TGAGCTCATTGAATGCTTGG	
M-TNF-α	F	ACCCTCACACTCACAAACCA	246
	R	GGCAGAGAGGAGGTTGACTT	
M-IL-17A	F	GACTACCTCAACCGTTCCAC	118
	R	CCTCCGCATTGACACAGC	
M-Muc-2	F	GACTGCCGAGACTCCTACAA	129
	R	CTTGTGGGTGAGGTAGATGG	
H-IL-8	F	TGGCTCTCTTGGCAGTC	238
	R	TGCACCCAGTTTTCCTTGGG	
H-TNF-α	F	AGCCCATGTTGTAGCAAACC	134
	R	TGAGGTACAGGCCCTCTGAT	
H-IL-1β	F	ATGATGGCTTATTACAGTGGCAA	132
	R	GTCGGAGATTCGTAGCTGGA	
H-IL-10	F	AAAGAAGGCATGCACAGCTC	132
	R	AAGCATGTTAGGCAGGTTGC	
H-GAPDH	F	CAACGACCACTTTGTCAAGC	140
	R	TTCCTCTTGTGCTCTTGCTG	

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
