# Peer review of "Lactobacillus reuteri Ameliorates Intestinal Inflammation and Modulates Gut Microbiota and Metabolic Disorders in Dextran Sulfate Sodium-Induced Colitis in Mice"

_nutrients, 2020, doi:10.3390/nu12082298_

Round 1

Reviewer 1 Report

Authors detail a comprehensive and well-structured study, obtaining promising results with high applicability due the GRAS condition of the bacterial species under study. It has been discussed in depth and clearly, and the conclusions are supported by the results.

However, some questions arise that I detail below:

  • Section 2.3

Is it a standard method, a new proposal one, a referenced method? In the latter case, please include the reference.

It would be interesting to include the LPS commercial product reference (LPS L ..., Sigma USA, E. coli) for reproducibility (There are many E.coli LPSs in that company).

How are HT-29 cells applied to wells? Is the antibiotic “penicillin” removed? Could it be negatively affecting L. reuteri I5007?

Has the LPS concentration after 4 hours incubation period with L. reuteri I5007 been analysed? Could there be degradation and/or modification of this?  Could that explain why the best results were found in the pre-treatment assay?

  • Section 2.4

In my opinion, the method could be clearer, for example, by following the structure in Section 2.3.

  • Section 2.6

Please detail the reaction mix and PCR conditions, or refer to a previously described method.

  • Section 2.9

How samples were prepared for UHPLC analysis? Please detail.

  • Section 3.6

The metabolism of some amino acids, e.g. histidine and tyrosine, can lead to the synthesis of biogenic amines, (in this example histamine and tyramine), which have potential to cause damage to the host. Lactic acid bacteria, and among them L. reuteri, are the main producers of these amines in food, which depends not only on the existence of the metabolic pathway, but also on the presence of precursors.

In this sense, it would be interesting to feed the mice with food of known composition (or to reflect it in the text if known) to estimate the true and/or potential safety and effects of the application of these bacteria, even more considering that the aforementioned routes exist, as can be deduced from Figure 5.C. It would also be very interesting to see a KEGG analysis comparing Control with L. reuteri I5007 treatment, as in the previous sections of the study.

  • Section 4

In the Discussion you say (Paragraph 6 of this section): “L. reuteri I5007 treatment dramatically decreased the abundance of Proteobacteria and increased that of Firmicutes” (Which is logical, since you are administering a member of the Firmicutes ...), and in line 14 of the same paragraph you indicate: “However, the administration of L. reuteri I5007 alone in the normal mice did not modulate microbiota composition, which was similar to the normal control group”. Is it a contradiction? Please explain.

In any case, the manuscript presented for review shows very interesting results on the possibility of using L. reuteri I5007 as a tool for the prevention and treatment of inflammatory bowel disease and associated pathologies, if some additional studies on security are carried in future.

Author Response

Response to Reviewer 1 Comments

  1. Point 1:

Section 2.3

(1) Is it a standard method, a new proposal one, a referenced method? In the latter case, please include the reference.

Response: Thanks for this comment. It is a referenced method. And we have added the reference in the revised manuscript.

(2) It would be interesting to include the LPS commercial product reference (LPS L ..., Sigma USA, E. coli) for reproducibility (There are many E.coli LPSs in that company).

Response: Thanks for this comment. We have added the detail in the revised manuscript.

(3) How are HT-29 cells applied to wells? Is the antibiotic “penicillin” removed? Could it be negatively affecting L. reuteri I5007?

Response: We are sorry we did not describe clearly. When cells reached 80 % confluency, the complete medium was removed completely, and cells were fed with fresh RPMI 1640 basic medium lacking antibiotics.

(4) Has the LPS concentration after 4 hours incubation period with L. reuteri I5007 been analysed? Could there be degradation and/or modification of this?  Could that explain why the best results were found in the pre-treatment assay?

Response: We are sorry we did not measure the concentration of LPS after 4h. LPS is a major component of the cell wall of Gram-negative bacteria and is very stable. As other inflammatory cell models induced by LPS, it is believed that the LPS concentration remains unchanged. From the results about the expression of inflammatory cytokines, we know that pretreatment effectively relieve the inflammation compare with the others.

  1. Point 2:

Section 2.4

In my opinion, the method could be clearer, for example, by following the structure in Section

Response: Thanks for this comment. We have supplemented the detail in this section.

  1. Point 3:

Section 2.6

Please detail the reaction mix and PCR conditions, or refer to a previously described method.

Response: Thanks for this comment. We have supplemented the detail about PCR analysis in this section.

  1. Point 4:

Section 2.9

How samples were prepared for UHPLC analysis? Please detail.

Response: We are sorry we did not describe clearly. We have supplemented the detail in this section.

  1. Point 5:

Section 3.6

The metabolism of some amino acids, e.g. histidine and tyrosine, can lead to the synthesis of biogenic amines, (in this example histamine and tyramine), which have potential to cause damage to the host. Lactic acid bacteria, and among them L. reuteri, are the main producers of these amines in food, which depends not only on the existence of the metabolic pathway, but also on the presence of precursors.

In this sense, it would be interesting to feed the mice with food of known composition (or to reflect it in the text if known) to estimate the true and/or potential safety and effects of the application of these bacteria, even more considering that the aforementioned routes exist, as can be deduced from Figure 5.C. It would also be very interesting to see a KEGG analysis comparing Control with L. reuteri I5007 treatment, as in the previous sections of the study.

Response: Yes, I agree that the function of these amines is two-sided and depends not only on the existence of the metabolic pathway, but also on the presence of precursors. As in this example, some bacteria could metabolite histidine to histamine, and the anti-inflammatory and pro-inflammatory effects of histamine depend on activated receptors (H2R or H1R).

  1. Point 6:

Section 4

(1) In the Discussion you say (Paragraph 6 of this section): “L. reuteri I5007 treatment dramatically decreased the abundance of Proteobacteria and increased that of Firmicutes” (Which is logical, since you are administering a member of the Firmicutes ...),

Response: Yes. L. reuteri is a member of the Firmicutes. In the present study, the relative abundance of Lactobacillus is less than 0.002%, but the relative abundance of Firmicutes increased by 7% after pretreatment of I5007. So the increased abundance of Firmicutes is not mainly determined by the abundance of added lactic acid bacteria.

(2) and in line 14 of the same paragraph you indicate: “However, the administration of L. reuteri I5007 alone in the normal mice did not modulate microbiota composition, which was similar to the normal control group”. Is it a contradiction? Please explain.

Response: No, it is not a contradiction. In healthy individuals, the intestinal microbiota is a stable ecosystem. Without strong external intervention (such as environmental factors, antibiotics and diet, etc.), the microbiota diversity will not change. Similar observations in randomized controlled trials (RCTs) of healthy adults reported that probiotics do not significantly modify the gut microbiota composition of healthy subjects.

Reviewer 2 Report

The paper : Lactobacillus reuteri Ameliorates Intestinal Inflammation and Modulates Gut Microbiota and Metabolic Disorders in Dextran Sulfate Sodium-Induced Colitis in Mice in my opinion is very well designed and written. However several issue should be improved: 

Abstract: please add DSS abreviation. It is first time mentioned.

Introduction: plese use the current definition of probiotic, according to Hill, C., Guarner, F., Reid, G., Gibson, G. R., Merenstein, D. J., Pot, B., ... & Calder, P. C. (2014). Expert consensus document: The International Scientific Association for Probiotics and Prebiotics consensus statement on the scope and appropriate use of the term probiotic. Nature reviews Gastroenterology & hepatology11(8), 506.

Several studies have shown that L. reuteri I5007 has several probiotic properties[20]. - The sentence is unclear, please add more details. Severeal studies ... - you should use at least two citation.

Results:

the figures (espetially Fig.1 and 3) are very small, I reccomend enlarge them

"L. reuteri I5007 administration greatly decelerated weight loss" - what you mean greatly? how much?

Discussion: when mention L. reuteri please add strain symbol. It is confusing when you discuss 15007 strain and when the other.

Author Response

Response to Reviewer2 Comments

  1. Point 1:

Abstract: please add DSS abreviation. It is first time mentioned.

Response: Thanks for this comment. We have supplemented the detail in this section.

  1. Point 2:

Introduction: plese use the current definition of probiotic, according to Hill, C., Guarner, F., Reid, G., Gibson, G. R., Merenstein, D. J., Pot, B., ... & Calder, P. C. (2014). Expert consensus document: The International Scientific Association for Probiotics and Prebiotics consensus statement on the scope and appropriate use of the term probiotic. Nature reviews Gastroenterology & hepatology, 11(8), 506.

Response: Thanks for this comment. We have added this reference in the revised manuscript.

  1. Point 3:

Several studies have shown that L. reuteri I5007 has several probiotic properties[20]. - The sentence is unclear, please add more details. Severeal studies ... - you should use at least two citation.

Response: We are sorry for our mistake. We have rewritten this sentence and added the references in the revised paper.

  1. Point 4:

Results:

(1) the figures (espetially Fig.1 and 3) are very small, I reccomend enlarge them

Response: Thanks for the valuable suggestion. We have modified the figures and replaced them.

(2) "L. reuteri I5007 administration greatly decelerated weight loss" - what you mean greatly? how much?

Response: Thanks for the comment. We have revised it in the new version.

(3) Discussion: when mention L. reuteri please add strain symbol. It is confusing when you discuss 15007 strain and when the other.

Response: Thanks for the comment. We have We have corrected it in the revised manuscript.

Reviewer 3 Report

Reviewer(s)' Comments to Author:

The authors describe an interesting study analyzing potential beneficial effects of probiotic supplementation with Lactobacillus reuteri on inflammatory, metabolic and gut microbiota markers related to DSS-induced colitis in mice. Interestingly, authors previously examined the in vitro effects of L. reuteri on inflammation in HT-29 cells, which has an added value since it allows establishing appropriate conditions for further in vivo tests. Methodology is well applied to achieve the proposed objectives. There is also scientific interest in the evaluation of those factors with critical roles in the pathogenesis of IBD. The paper is generally well-written.

However, I have some questions and concerns about the manuscript. 

Authors make extensive use of terms "administration", "supplementation" or "treatment", especially in discussion section. I believe that it is more appropriate to use "pre-treatment", which corresponds more closely to the experimental conditions defined in this study (see Figure 1 and Figure 2A). This should be reviewed by authors, as far as possible.

In ABSTRACT, it would be preferable to define the term DSS as dextran sulphate sodium

I would like to suggest that abbreviations not be used as keywords (DSS, IBD), and they are replaced by their full definition.

INTRODUCTION supports background information necessary to provide a specific context for the results, explaining both the biological value of the variables to be analyzed in IBD pathology as well as potential beneficial effects of L. reuteri as probiotic strain. Specific objectives are clearly defined.

MATERIALS AND METHODS are clearly explained, providing detailed information about cell culture conditions, in vitro procedures, treatments and methodologies developed to evaluate the potential beneficial effects of probiotic pre-treatment in mice with DSS-induced colitis.

Specific comments:

  1. Section Cell culture and treatment. Please, indicate the concentrations of L. reuteri and LPS used in pre-, co- and post-treatment to better reflect that all experimental conditions are performed under the same concentrations
  2. Section Animals and Treatments. Please, indicate the concentration of DSS used to induce colitis. Authors should indicate experimental groups as well as the number of mice included for each experimental group. Figure 2A should be included in this section to better explain methodology used.
  3. Section Histological evaluation. Please insert information about light microscopy (model) used in the study.
  4. Section Quantitative Real-Time Polymerase Chain Reaction (PCR) Analysis. Please, complete method with information about system used for qRT-PCR, PCR program conditions and software used for data analysis.
  5. Section Bioinformatics analysis of 16S rRNA gene profiling. All abbreviations used in this section should be defined (and not include them in results section).
  6. Section Statistical analysis. “Dunn´s multiple comparison test in case of abnormal distribution” should be replaced by non-normal or non-parametric distribution.

Some suggestions for changes in RESULTS are discussed below.

3.1 Effects of L. reuteri I5007 on inflammatory cytokines in HT-29 cells challenged with LPS.

Levels of mRNA expression of cytokines analyzed are different in the same experimental group (Control, LPS, I5007) when analyzing under the three experimental conditions (pre-, co-, post-treatment). For example, IL-8 mRNA level is higher in LPS group under co-treatment conditions (mRNA relative expression ⁓8) than LPS group under post-treatment conditions (mRNA relative expression ⁓4.5). Due to both L-reuteri and LPS concentrations are similar in the different experimental conditions, how do you explain these results?

3.2 Effects of L. reuteri I5007 on colitis symptoms. As noted above, Fig2A should be included in Methods section.

  • Authors support that “Untreated and L. reuteri I5007-treated mice continuously gained weight”; however, data showed in Fig 2B seem to indicate that body weight challenge remains constant in both groups throughout time. This sentence should be reviewed.
  • Please, experimental groups should maintain in the same order in all figures.
  • In figure 2D, is it possible to indicate histological changes you describe?
  • Please, indicate P<0.0001 (****) in figure footnotes.

3.3 Effects of L. reuteri I5007 on cytokines expression in colon tissue and serum.

Why didn't you analyze IL-1beta levels or IL-10 expression? It is important to evaluate simultaneously data from gene expression and levels of the same pro- and anti-inflammatory cytokines. Moreover, results obtained for IL-10 levels should be described in text.

3.4 Effects of L. reuteri I5007 on microbiota composition in colon contents.

According to Fig 4C, relative abundance of Bacteriodetes seems to increase in DSS-I5007 groups compared to DSS-group. If this is true, it should be included in the results obtained.

3.6 Effects of L. reuteri I5007 metabolite composition in colonic contents

Metabolites in the colonic contents were evaluated between control vs DSS, and DSS vs DSS-I5007. Were comparison control vs DSS-I5007 performed? It would be important to analyze whether metabolites in colonic contents of DSS-I5007 group is close or not to control group.

Regarding to DISCUSSION section,

  • First paragraph: minor grammatical errors should be corrected: supplemtnation, numers and matintained.
  • Second paragraph: This sentence should be rewritten “In the present study, we found that   reuteri  I5007  effectively  mitigated  body  weight  loss  and  increased  colon  length  as  well  as pathological changes in the colon in DSS-induced colitis mice” in order to clarify the effects of L. reuteri I5007 on pathological changes in DSS-induced colitis mice.
  • Third paragraph: “TNF-α is an activator of NF-κB, which further promotes the secretion of TNF-α and upregulates other proinflammatory cytokines, such as IL-1β and IL-6”. In fact, TNF-alpha is an activator of NF-kB y, but other pro-inflammatory cytokines, such as IL-1beta, also activate this pro-inflammatory signaling pathway (see ref Lawrence T. The nuclear factor NF-kappaB pathway in inflammation. Cold Spring Harb Perspect Biol. 2009;1(6):a001651. doi:10.1101/cshperspect.a001651 as example). This sentence should be rewritten. Moreover, results obtained from IL-10 analysis should be discussed. It is important to evaluate both pro- and anti-inflammatory effects of reuteri pre-treatment.
  • Sixth paragraph: Authors suggest that “L. reuteri I5007 treatment dramatically decreased the abundance of Proteobacteria and increased that of Firmicutes, which were reported to be associated with IBD development”. This sentence seems to indicate that changes in intestinal microbiota induced by reuteri are similar to those caused by pathology, so it would be a negative effect of treatment. Authors should clarify this point.
  • Following on from my previous point, in seventh paragraph, how do IBD development and gut microbiota dysbiosis affect metabolism of amino acids? Are there differences in amino acids metabolism between IBD and reuteri pre-treatment?
  • Finally, (eighth paragraph), conclusions about antioxidant effects of L-reuteri treatment should be moderated due to authors did not evaluate antioxidant systems in this study (mRNA levels of antioxidant enzymes, intracellular antioxidant levels, or enzymatic activity), and their conclusions are largely based on studies by other authors.

Author Response

Response to Reviewer3 Comments

  1. Point 1:

Authors make extensive use of terms "administration", "supplementation" or "treatment", especially in discussion section. I believe that it is more appropriate to use "pre-treatment", which corresponds more closely to the experimental conditions defined in this study (see Figure 1 and Figure 2A). This should be reviewed by authors, as far as possible.

Response: Thanks for the valuable suggestion. We have made the changes in the revised manuscript.

  1. Point 2:

In ABSTRACT, it would be preferable to define the term DSS as dextran sulphate sodium

I would like to suggest that abbreviations not be used as keywords (DSS, IBD), and they are replaced by their full definition.

Response: Thanks for this comment. We have added the detail in the new paper.

  1. Point 3:

Section Cell culture and treatment. Please, indicate the concentrations of L. reuteri and LPS used in pre-, co- and post-treatment to better reflect that all experimental conditions are performed under the same concentrations

Section Animals and Treatments. Please, indicate the concentration of DSS used to induce colitis. Authors should indicate experimental groups as well as the number of mice included for each experimental group. Figure 2A should be included in this section to better explain methodology used.

Section Histological evaluation. Please insert information about light microscopy (model) used in the study.

Section Quantitative Real-Time Polymerase Chain Reaction (PCR) Analysis. Please, complete method with information about system used for qRT-PCR, PCR program conditions and software used for data analysis.

Section Bioinformatics analysis of 16S rRNA gene profiling. All abbreviations used in this section should be defined (and not include them in results section).

Section Statistical analysis. “Dunn´s multiple comparison test in case of abnormal distribution”should be replaced by non-normal or non-parametric distribution.

Response: We are sorry we did not describe clearly. We have added the detail in the revised manuscript.

  1. Point 4:

(1) 3.1 Effects of L. reuteri I5007 on inflammatory cytokines in HT-29 cells challenged with LPS.

Levels of mRNA expression of cytokines analyzed are different in the same experimental group (Control, LPS, I5007) when analyzing under the three experimental conditions (pre-, co-, post-treatment). For example, IL-8 mRNA level is higher in LPS group under co-treatment conditions (mRNA relative expression ⁓8) than LPS group under post-treatment conditions (mRNA relative expression ⁓4.5). Due to both L-reuteri and LPS concentrations are similar in the different experimental conditions, how do you explain these results?

Response: We have checked the data of IL-8 mRNA level under co-treatment and post-conditions. Their expressions are similar. We did not adjust the level in the control group to 1. Sorry for taking the confusion.

(2) Authors support that “Untreated and L. reuteri I5007-treated mice continuously gained weight”; however, data showed in Fig 2B seem to indicate that body weight challenge remains constant in both groups throughout time. This sentence should be reviewed.

Response: Thanks for this comment. We have rewritten this sentence in the revised manuscript.

(3) Please, experimental groups should maintain in the same order in all figures.

Response: Thanks for the comment. We have changed the figures in the new version.

(4) In figure 2D, is it possible to indicate histological changes you describe?

Response: Yes. From HE staining of colon tissue, we can see clearly colonic histopathology changes after DSS inducing. And pretreatment of L. reuteri I5007 decreased mucosal ulceration, epithelial edema and crypt loss.

(5) Please, indicate P<0.0001 (****) in figure footnotes.

Response: We have added it in the revised manuscript.

(6) Why didn't you analyze IL-1beta levels or IL-10 expression? It is important to evaluate simultaneously data from gene expression and levels of the same pro- and anti-inflammatory cytokines. Moreover, results obtained for IL-10 levels should be described in text.

Response: We did measure IL-10 mRNA level, but there was no difference. The aim of measuring cytokine concentration is to clarify that I5007 can relieve inflammation. As we descripted in the method, we measured the cytokines in the serum using Mouse Th17 Panel (8-plex) array including 8 cytokines. The changes of these cytokines can indicate the effect of I5007 in regulating the inflammation. I agree that it is important to evaluate simultaneously data from gene expression and levels of the same pro- and anti-inflammatory cytokines. Thanks for the valuable suggestion.

       We also add the description about IL-10 levels in the text.

(7) According to Fig 4C, relative abundance of Bacteriodetes seems to increase in DSS-I5007 groups compared to DSS-group. If this is true, it should be included in the results obtained.

Response: Thanks for the comment. We have changed the description in this section.

Metabolites in the colonic contents were evaluated between control vs DSS, and DSS vs DSS-I5007. Were comparison control vs DSS-I5007 performed? It would be important to analyze whether metabolites in colonic contents of DSS-I5007 group is close or not to control group.

Response: We did not compare control vs DSS_I5007. We think they are not comparable. Because they don’t have the same background, the only variable. Control group is the normal mice, and DSS_I5007 group is the colitis mice. Depend on the comparison DSS vs DSS_I5007, we can indicate that pretreatment of I5007 have a modulation of metabolite composition. We also did compare Control vs I5007, we find that the composition between these two groups was similar.

  1. Point 5:

(1) First paragraph: minor grammatical errors should be corrected: supplemtnation, numers and matintained.

Second paragraph: This sentence should be rewritten “In the present study, we found that   reuteri  I5007  effectively  mitigated  body  weight  loss  and  increased  colon  length  as  well  as pathological changes in the colon in DSS-induced colitis mice” in order to clarify the effects of L. reuteri I5007 on pathological changes in DSS-induced colitis mice.

Third paragraph: “TNF-α is an activator of NF-κB, which further promotes the secretion of TNF-α and upregulates other proinflammatory cytokines, such as IL-1β and IL-6”. In fact, TNF-alpha is an activator of NF-kB y, but other pro-inflammatory cytokines, such as IL-1beta, also activate this pro-inflammatory signaling pathway (see ref Lawrence T. The nuclear factor NF-kappaB pathway in inflammation. Cold Spring Harb Perspect Biol. 2009;1(6):a001651. doi:10.1101/cshperspect.a001651 as example). This sentence should be rewritten. Moreover, results obtained from IL-10 analysis should be discussed. It is important to evaluate both pro- and anti-inflammatory effects of reuteri pre-treatment.

Response: Thanks for the valuable suggestion. We have rewritten these sentences in the revised manuscript according to the comment.

(2) Sixth paragraph: Authors suggest that “L. reuteri I5007 treatment dramatically decreased the abundance of Proteobacteria and increased that of Firmicutes, which were reported to be associated with IBD development”. This sentence seems to indicate that changes in intestinal microbiota induced by reuteri are similar to those caused by pathology, so it would be a negative effect of treatment. Authors should clarify this point.

Response: The increased abundance of Proteobacteria may increase the risk of IBD, but not the direct reason. We have corrected this ambiguous sentence in the revised manuscript.

(3) Following on from my previous point, in seventh paragraph, how do IBD development and gut microbiota dysbiosis affect metabolism of amino acids? Are there differences in amino acids metabolism between IBD and reuteri pre-treatment?

Response: Yes. Amino acids are not only absorbed in the small intestine, but also metabolized and utilized by microbiota in the large intestine. The amino acids can be metabolized to various metabolites, such as indole acids, polyamines and histamine. Much research has showed that these metabolites have a modulation on immunity. In the present study, we found that the differences in amino acids metabolism and concentration.

(4) Finally, (eighth paragraph), conclusions about antioxidant effects of L-reuteri treatment should be moderated due to authors did not evaluate antioxidant systems in this study (mRNA levels of antioxidant enzymes, intracellular antioxidant levels, or enzymatic activity), and their conclusions are largely based on studies by other authors

Response: Thanks for this comment. We did not further verify the findings, and they are what we would further study to clarify the mechanism.

Reviewer 4 Report

The project is essentially an extension of an earlier rat study of L. reuterii/fermentum in DSS-colitis by Geier et al. However, this vital reference appears to have been overlooked by the authors. MS Geier et al. Lactobacillus fermentum BR11, a potential new probiotic, alleviates symptoms of colitis induced by dextran sulfate sodium (DSS) in rats. Int J Food Micro 114(3):267-274 (2007).

English grammar requires significant attention throughout the manuscript.. Examples in the abstract include 'many various'.... 'relieved inflammation in HT-29 cells'...and 'mice were given with'.

The introduction is lacking specific aims and hypotheses. Numbers of mice/group are not stated in Methods. Figures are too small to read; especially the histology sections. Actual results are well described. Grammatical errors in most sentences make readability difficult. The word 'supplementation' is misspelt in the first line of the Discussion.

Author Response

Response to Reviewer4 Comments

  1. Point 1:

The project is essentially an extension of an earlier rat study of L. reuterii/fermentum in DSS-colitis by Geier et al. However, this vital reference appears to have been overlooked by the authors. MS Geier et al. Lactobacillus fermentum BR11, a potential new probiotic, alleviates symptoms of colitis induced by dextran sulfate sodium (DSS) in rats. Int J Food Micro 114(3):267-274 (2007).

Response: Thanks for the comment. We have added this reference in the revised manuscript.

  1. Point 2:

English grammar requires significant attention throughout the manuscript.. Examples in the abstract include 'many various'.... 'relieved inflammation in HT-29 cells'...and 'mice were given with'.

Response: About the English grammar of the manuscript, we have asked for native English speaker to check the paper.

  3. Point 3:

The introduction is lacking specific aims and hypotheses. Numbers of mice/group are not stated in Methods. Figures are too small to read; especially the histology sections. Actual results are well described. Grammatical errors in most sentences make readability difficult. The word 'supplementation' is misspelt in the first line of the Discussion.

Response: Thanks for these valuable suggestions. We have rewritten these sentences in the revised manuscript according to the comments.
